# Band Polarization Effect on the Kondo State in a Zigzag Silicene Nanoribbon

**DOI:** 10.3390/nano12091480

**Published:** 2022-04-27

**Authors:** Ginetom S. Diniz, Edson Vernek, George B. Martins

**Affiliations:** 1Curso de Física, Universidade Federal de Jataí, Jataí 75801-615, GO, Brazil; ginetom@gmail.com; 2Instituto de Física, Universidade Federal de Uberlândia, Uberlândia 38400-902, MG, Brazil; vernek@ufu.br

**Keywords:** Kondo effect, topological insulators, silicene

## Abstract

Using the Numerical Renormalization Group method, we study the properties of a quantum impurity coupled to a zigzag silicene nanoribbon (ZSNR) that is subjected to the action of a magnetic field applied in a generic direction. We propose a simulation of what a scanning tunneling microscope will see when investigating the Kondo peak of a magnetic impurity coupled to the metallic edge of this topologically non-trivial nanoribbon. This system is subjected to an external magnetic field that polarizes the host much more strongly than the impurity. Thus, we are indirectly analyzing the ZSNR polarization through the STM analysis of the fate of the Kondo state subjected to the influence of the polarized conduction electron band. Our numerical simulations demonstrate that the spin-orbit-coupling-generated band polarization anisotropy is strong enough to have a qualitative effect on the Kondo peak for magnetic fields applied along different directions, suggesting that this contrast could be experimentally detected.

## 1. Introduction

Topological insulators have attracted much attention since the proposal by Charles Kane and Eugene Mele [1,2] was made, stating that the spin-orbit interaction in graphene could reproduce the properties of the seminal model proposed by Duncan Haldane in 1988 [3], which displayed topological properties similar to those of the quantum Hall effect; however, without a net magnetic field. The topological effect in the so-called Kane–Mele model, which was supposed to display ‘spin-momentum-locked’ edge states at the borders of a graphene finite-size sample, was dubbed the ‘Quantum Spin Hall Effect’. Unfortunately, graphene’s spin-orbit interaction (essential for the opening of a gap and the spin-momentum-locking effect) proved to be vanishingly small, and this non-trivial topological state had to be pursued in an specific quantum-well system, CdTe/HgTe [4,5]. However, it so happens that the so-called *Xenes* (silicene, germanene and stanene, among others), which form monolayers that share many similarities with graphene, have a spin-orbit interaction that is three orders of magnitude larger than graphene [6]. This happens because of an effect called ‘buckling’, where, contrary to graphene, the two sublattices forming the Xenes’ hexagonal structure are not coplanar, rather, they are shifted vertically. Thus, Xenes can be very closely described by the Kane–Mele model, and therefore a zigzag silicene nanoribbon (ZSNR), for example, will present a 1D helical state at its edges: counter-propagating spin-momentum-locked states that form what became known as a helical 1D liquid. Reviews of Xenes properties can be found in Refs. [7,8,9,10,11,12,13,14,15]. Investigations treating more specifically the topological properties of silicene, germanene, and stanene are found in Refs. [16,17]. A detailed description of how to obtain the Kane–Mele model as a low energy effective model for silicene, germanene, and stanene, starting from first principles calculations, can be found in Ref. [6].

Given that the helical edge state is protected by time-reversal symmetry [1,2], and is thus immune, in principle, to potential scattering, it is interesting to ask what the influence is on the helical state of magnetic impurities that couple to it. Indeed, due to the helical state property of spin-momentum locking, back-scattering has to involve a spin-flip; thus, it cannot occur for pure potential scattering, while a magnetic impurity, which can flip its spin too, would provide a back-scattering mechanism. For example, the fact that the conductance of the helical state in CdTe/HgTe was measured to be $G<G_{0}=2e^{2}\;/\;h$ and decreased with lowering temperature, was attributed, at least initially [18,19], to the presence of either magnetic impurities coupled to the edges or to the presence of trapped electrons that interact with the helical state [20]. It should be noticed that, recently, an alternative interpretation to the conductance measurements in CdTe/HgTe quantum-wells have been proposed, see Refs. [18,19]. In both cases, this can be modeled by a single-impurity Anderson model (SIAM) [21], and points to the possible occurrence of a Kondo effect [22]. For a detailed discussion of the various mechanisms invoked to explain the $G<G_{0}$ anomalous conductance, see Ref. [23].

Few studies have directly addressed the Kondo effect for the case of a quantum magnetic impurity coupled to the edge state associated with the Kane–Mele model. First, Goth et al. [24] have used the continuous time quantum Monte Carlo method to study the spatial dependence of the single-particle spectral functions and spin–spin correlation functions when a single quantum impurity couples to the helical edge state. They concluded that, contrary to what happens for a one-dimensional conductor, in a helical liquid, magnetic impurities cannot block transport below the Kondo temperature. Rather, the current circumvents the impurity. A similar result was obtained by Allerdt et al. [25,26], using a recently developed numerical method combined with the Density Matrix Renormalization Group method [27,28]. It should also be noted that Weymann et al. [29], using the Numerical Renormalization Group (NRG) method, coupled to the Density Functional Theory, have conducted a detailed study of the Kondo effect of a cobalt impurity deposited on top of a 2D silicene sheet. As the host for their impurity has no edges, just bulk, they did not analyze any of the properties of the edge state in relation to Kondo coupling. However, they did perform a careful analysis of the effects of gating (variation of the band filling of the host), as well as the effect of an external magnetic field acting just on the cobalt impurity.

In the present work, using NRG, we intend to expand the research conducted on the Kondo effect stemming from a quantum impurity coupled to a ZSNR. We study the effect of a magnetic field applied upon the host, not just upon the impurity. It is known that the magnetic polarization of the bands, due to an external magnetic field, affects the Kondo state [30], mainly if the host displays spin-orbit interaction, as the band magnetic polarization becomes anisotropic. Since there is ample choice of what kind of magnetic impurity may be adsorbed into the ZSNR, demonstrating a wide variation in their g-factor, we will explicitly consider a situation where the host g-factor, denoted $g_{b}$ (see Equation (Equation 2)), is considerably larger than the impurity g-factor, denoted $g_{\mathrm{imp}}$ (see Equation (Equation 3)). This is intended to showcase the effect on the Kondo state caused by the band magnetic polarization.

## 2. Theoretical Model

### 2.1. Model Hamiltonian

To model a magnetic impurity coupled to a ZSNR, as depicted in Figure 1, we use a SIAM-like Hamiltonian [21], given by
(1)$$\begin{array}{c} H\;=\;H_{\mathrm{ZSNR}}+H_{\mathrm{imp}}+H_{\mathrm{tip}}+H_{\mathrm{ZSNR}-\mathrm{imp}}+H_{\mathrm{imp}-\mathrm{tip}}, \end{array}$$
where the first term describes a ZSNR, subjected to an external magnetic field and modeled by a tight-binding Hamiltonian, which in real space reads as
(2)$$\begin{array}{ccc} H_{\mathrm{ZSNR}} & = & \sum_{i\sigma }\varepsilon_{0}c_{i\sigma }^{†}c_{i\sigma }-\left[t\sum_{\langle i,j\rangle \sigma }c_{i\sigma }^{†}c_{j\sigma }-i\frac{\lambda_{\mathrm{SO}}}{3\sqrt{3}}\sum_{\langle \;\langle i,j\rangle \;\rangle \sigma }\sigma \nu_{ij}c_{i\sigma }^{†}c_{j\sigma }+H.c.\right]\\ & + & \mu_{B}g_{b}\sum_{i}{\vec{S}}_{i}\cdot \vec{B}. \end{array}$$

The first term in Equation (Equation 2) is the on-site energy, where the operator $c_{i\sigma }^{†}$ ($c_{i\sigma }$) creates (annihilates) an electron with energy $\varepsilon_{0}$ and spin σ in the *i*-th site of the ZSNR. The second term is the nearest-neighbor (NN) π-band tight-binding term, where $t_{ij}=t$ is the hopping between NN sites. The introduction of a next-NN hopping will not qualitatively alter the results, since the presence of the impurity in a hollow-site configuration already breaks particle-hole symmetry, see Ref. [31] for further details. The third term is the intrinsic next-NN spin-orbit coupling (SOC) $\lambda_{\mathrm{SO}}$, where $\nu_{ij}=+1$ if the NNN hopping is anticlockwise and $\nu_{ij}=-1$ if it is clockwise (in relation to the positive *z*-axis) [6]. Note that, in the third term, σ=± for subindex σ=↑↓, respectively. The fourth term is the Zeeman interaction due to an external magnetic field $\vec{B}=\left(B_{x},B_{y},B_{z}\right)$, where $\mu_{B}$ stands for the Bohr magneton, $g_{b}$ is the conduction electron *g*-factor for the ZSNR, and ${\vec{S}}_{i}$ is the conduction electron spin density in site *i*. Finally, all calculations are conducted with the ZSNR at half-filling and we use the NN hopping *t* as our unit of energy. It is worth mentioning that the SOC term is equivalent to an effective magnetic field perpendicular to the ZSNR (z-direction). This is discussed in Appendix B.

The second term in Equation (Equation 1) describes the impurity, including a Zeeman interaction,
(3)$$\begin{array}{ccc} H_{\mathrm{imp}} & = & \epsilon_{d}\left(n_{\mathrm{imp},↑}+n_{\mathrm{imp},↓}\right)+Un_{\mathrm{imp},↑}n_{\mathrm{imp},↓}+\mu_{B}g_{\mathrm{imp}}{\vec{S}}_{d}\cdot \vec{B}, \end{array}$$
where $n_{\mathrm{imp},\sigma }=d_{\sigma }^{†}d_{\sigma }$ and $d_{\sigma }^{†}$ ($d_{\sigma }$) creates (annihilates) an electron at the impurity with orbital energy $\epsilon_{d}$, while double occupancy of the impurity costs Coulomb energy *U*. In the third (Zeeman) term, $g_{\mathrm{imp}}$ stands for the impurity *g*-factor and ${\vec{S}}_{d}$ is the impurity spin operator.

As most experimental setups that are employed to probe such systems count with the aid of a scanning tunneling microscope (STM) device, we have also added a third term to Equation (Equation 1), referring to an STM tip, which is modeled by the Hamiltonian $H_{\mathrm{tip}}=\sum_{M,k,\sigma }\varepsilon_{Mk}c_{Mk\sigma }^{†}c_{Mk\sigma }$, where $c_{Mk\sigma }^{†}$ ($c_{Mk\sigma }$) creates (annihilates) an electron with momentum *k* and spin σ in the metallic STM tip. The fourth term of Equation (Equation 1), which couples the impurity to the ZSNR, is given by
(4)$$\begin{array}{c} H_{\mathrm{ZSNR}-\mathrm{imp}}=\sum_{j,\sigma }\left(V_{j\sigma }c_{j\sigma }^{†}d_{\sigma }+H.c.\right), \end{array}$$
while the fifth term, which couples the impurity to the STM tip, reads as
(5)$$\begin{array}{c} H_{\mathrm{imp}-\mathrm{tip}}=\sum_{k,\sigma }\left(V_{k}c_{k,\sigma }^{†}d_{\sigma }+H.c.\right). \end{array}$$

In Equation (Equation 4), $V_{j\sigma }$ represents the impurity-ZSNR hopping amplitude to the six neighboring silicon atoms when the impurity is in the hollow-site configuration (see Figure 1). Assuming the same coupling strength to the impurity for all the nearest-neighbor Si atoms, we will set $V_{j\sigma }\equiv V_{0}$. We have checked that considering the influence of the buckling in the values of $V_{j\sigma }$, i.e., having different hoppings from the impurity to different sublattice-sites, does not qualitatively change the results obtained. The value we used for the sublattice vertical shift was 0.46 Å [17]. In addition, assuming a constant density of states at the tip, $\rho_{tip}$, and taking $V_{k}\equiv V_{tip}$ in Equation (Equation 5), we may write the tip hybridization function as $\Gamma_{tip}=\pi V_{tip}^{2}\rho_{tip}$. Based on the results presented by Weymann et al. [29], the energetically most favorable position for a cobalt impurity in silicene is the so-called hollow-site configuration (depicted in Figure 1). We will thus present all our results for this impurity-coupling configuration. Finally, to keep the analysis simple, we will neglect the Rashba SOC, so that the only terms that control the band gap are the external magnetic field and the intrinsic SOC, $\lambda_{SO}$.

### 2.2. Tight-Binding Bands

The tight-binding ZSNR Hamiltonian (Equation (Equation 2)) can be diagonalized by a Fourier transform to reciprocal space, resulting in
(6)$$\begin{array}{cc} H_{\mathrm{ZSNR}} & =\sum_{k}\Psi_{k}^{†}H_{\mathrm{ZSNR}}\Psi_{k}, \end{array}$$
where $\Psi_{k}^{†}=\left(c_{k↑}^{†},c_{k↓}^{†}\right)$, $c_{k\sigma }^{†}$ creates an electron with wave vector *k* and spin σ=↑,↓, and the spectra is obtained by numerical diagonalization. In Figure 2, we show the resulting band structure (left panels) and total density of states (DOS, right panels) for a ZSNR with N=26 zigzag chains. The top panels have results for the vanishing magnetic field, with $\lambda_{SO}=0.0$ (black curve) and $\lambda_{SO}=0.1$ (red curve). In Ref. [6], the $\lambda_{SO}$ estimates for silicene, germanene, and stanene are (in meV), respectively, 3.9, 43.0, and 29.9. Thus, we are assuming a relatively high-value for $\lambda_{SO}$ (more appropriate for germanene); nonetheless, the qualitative behavior of the results presented in the following sections still holds for the case of a lower $\lambda_{SO}$. In the absence of intrinsic SOC, the well-known dispersionless bands at the Fermi energy, associated to the edge states present in a graphene ZNR [32], can be observed in Figure 2a (black curve), with a noticeable peak in the respective DOS, Figure 2b. The introduction of the intrinsic SOC adds dispersion to these metallic bands (red curve), besides polarizing the edge state spins [17]. When a magnetic field is applied to the ZSNR, as shown in Figure 2c, there is a spin splitting proportional to the Zeeman energy. As a consequence, the dispersionless $\lambda_{SO}=0.0$ energy bands at the Fermi level (black curve) are now symmetrically (particle-hole) shifted, but still present sharp singularities in the DOS, which are only suppressed for $\lambda_{SO}\neq 0$ (red curve), as shown in Figure 2d. In the next sections, we will demonstrate that these changes in the DOS have interesting consequences to the Kondo effect.

### 2.3. Hybridization Function

The impurity Green’s function ${\hat{G}}_{\mathrm{imp}}\left(k,\omega \right)$ can be written as
(7)$$\begin{array}{c} {\hat{G}}_{\mathrm{imp}}\left(k,\omega \right)={\left[\left(\omega -\epsilon_{d}\right)\sigma_{0}-{\hat{\Sigma }}^{\left(\mathrm{int}\right)}\left(k,\omega \right)-{\hat{\Sigma }}^{\left(0\right)}\left(k,\omega \right)\right]}^{-1}, \end{array}$$
where ${\hat{\Sigma }}^{\left(\mathrm{int}\right)}\left(k,\omega \right)$ is the interaction self-energy, while ${\hat{\Sigma }}^{\left(0\right)}\left(k,\omega \right)=\hat{V}{\hat{G}}_{\mathrm{ZSNR}}\left(k,\omega \right){\hat{V}}^{†}$ is the hybridization self-energy, with $\hat{V}=V_{0}\sigma_{0}$ and ${\hat{G}}_{\mathrm{ZSNR}}\left(k,\omega \right)={\left[\omega \sigma_{0}-H_{\mathrm{ZSNR}}\right]}^{-1}$. For a magnetic field applied along an arbitrary direction, ${\hat{\Sigma }}^{\left(0\right)}\left(k,\omega \right)$ has finite off-diagonal terms and we have to deal with a spin-mixing hybridization function [33,34]
(8)$$\begin{array}{c} \hat{\Gamma }\left(\omega \right)=\frac{1}{2i}\int_{-\pi }^{\pi }\left[{\hat{\Sigma }}^{\left(0\right)}\left(k,\omega -i0^{-}\right)-{\hat{\Sigma }}^{\left(0\right)}\left(k,\omega +i0^{+}\right)\right]dk. \end{array}$$

This positive-definite Hermitian matrix can be decomposed in terms of Pauli matrices as $\hat{\Gamma }\left(\omega \right)=\sum_{\alpha =0,x,y,z}d_{\alpha }\left(\omega \right)\sigma_{\alpha }$, where all $d_{\alpha }\left(\omega \right)$ are real quantities. In particular, $d_{0}\left(\omega \right)$ is proportional to the conduction-band density of states. More specifically, the spin-resolved density of states is given by $\rho_{\sigma \sigma }\left(\omega \right)=\Gamma_{\sigma \sigma }\left(\omega \right)/\left(\pi V^{2}\right)$. Thus, $\rho_{↑↑}\propto d_{0}+d_{z}$, $\rho_{↓↓}\propto d_{0}-d_{z}$, resulting in $\rho =\rho_{↑↑}+\rho_{↓↓}\propto 2d_{0}$.

In Figure 3, we show, in the main panels, the coefficients $d_{\alpha }\left(\omega \right)$ of the hybridization function decomposition, in terms of the Pauli matrices $\sigma_{\alpha }$, for the vanishing SOC. Each panel contains a different magnetic field configuration: it vanishes in panel (a), and it takes values $B_{\alpha }=0.001$, for α=x,y,z in panels (b) to (d), respectively. The coefficients $d_{\alpha }$ contain important physical information about the system under study. For instance, in panel (a), in the absence of any interaction (vanishing SOC and magnetic field), the system presents a localized edge state at the Fermi level, as shown in Figure 2b, whose signature is captured by the $d_{0}$ coefficient, which is directly proportional to the ZSNR DOS (see above), while all the other coefficients vanish. For the case of finite magnetic field [panels (b) to (d)] the $d_{0}$ peak observed in panel (a) is spin-split due to the Zeeman interaction. In addition, for each field orientation $B_{\alpha }$, besides $d_{0}$, only the corresponding coefficient $d_{\alpha }$ will be finite, although its dependence with ω is the same for all three directions, as can be observed in Figure 3b–d. This is a consequence of SU(2) symmetry (spatial isotropy of the conduction electron spins) in the absence of SOC.

If the intrinsic SOC is turned on, i.e., $\lambda_{SO}=0.1$, first, as implied from Figure 2b (red curve), the peak in $d_{0}$ at the Fermi energy should be strongly suppressed and broadened, as shown in the inset to Figure 3a. In addition, the spatial isotropy observed for the vanishing SOC is broken. Indeed, as shown in the Appendix B, the intrinsic SOC may be interpreted as a *k*-dependent effective magnetic field along the *z*-axis. Thus, the results for magnetic field along the *x*- or *y*-axis should be identical (with *y* interchanged for *x*, see the insets in Figure 3b,c), but they should differ from the results for the magnetic field along the *z*-axis (see inset in Figure 3d). It is also important to mention that even though the ZSNR preserves particle-hole (*p*-*h*) symmetry when SOC and magnetic fields are considered, for the impurity located at the hollow site, the system is non-bipartite; therefore, the hybridization function does not preserve particle-hole symmetry (*p*-*h* symmetry) [35]. However, the broken *p*-*h* symmetry is highly pronounced only for high energies (not shown in the figures).

Before presenting the NRG results, we wish to discuss the results for the ZSNR band polarization $\langle S_{\alpha }^{b}\rangle$, which is calculated as $\sum_{k}\langle k|S_{\alpha }^{b}|k\rangle$, where |k〉 are the eigenstates of $H_{\mathrm{ZSNR}}$ in Equation (Equation 6). The sum is over the first Brillouin zone, up to the Fermi energy. Figure 4 shows the results for vanishing SOC (black curve), when all magnetic field orientations produce the same result, and for $\lambda_{SO}=0.1$, where the $\langle S_{x/y}^{b}\rangle$ for a magnetic field applied along the *x* or *y*-axis are identical (see red and green curves), while the polarization $\langle S_{z}^{b}\rangle$ for a magnetic field applied along the *z*-axis is suppressed (blue curve). The anisotropy observed in Figure 4 was already discussed in connection with Figure 3, being caused by the SOC effective magnetic field oriented along the *z*-axis. As shown in the next section, the considerably smaller polarization at finite SOC for the *z*-axis when compared to the *x*- and *y*-axis, has sizable consequences for the Kondo effect when $g_{\mathrm{imp}}=0$, which would be experimentally detectable for a real situation where the magnetic impurity is such that $g_{\mathrm{imp}}\ll g_{b}$ (this subject is discussed in detail in Appendix A).

In the following section, we discuss the Kondo effect when a magnetic impurity is coupled to the edge of a ZSNR (see Figure 1). To do so, we perform NRG calculations to obtain the spectral function and thermodynamic properties of this system. To this end, we use the well-known NRG Ljubljana open source code [36]. For all calculations, we have used the discretization parameter Λ=2.0 and kept 2000 states at each iteration. We also employ the z-trick [37] (with z=0.2, 0.4, 0.6, 0.8 and 1.0) to remove unwanted oscillations in the physical quantities. To map our 2 × 2 matrix hybridization function onto a Wilson chain we have employed the scheme proposed by Liu et al. in Ref. [33]

### 2.4. Results and Discussion

To obtain the following numerical results, we have considered N=26, $\Gamma_{tip}=0.01$, U=0.5, and $V_{0}=0.05$ for all figures, while different sets of values for $\lambda_{SO}$ and $\vec{B}$ were used. As mentioned in the Introduction, we also considered $g_{\mathrm{imp}}=0$ (with $g_{b}=1$) in all calculations.

In Figure 5 we show how the impurity density of states ρ(ω), at zero magnetic field (black curve in both panels), is affected by the band magnetic polarization for a magnetic field applied along $\hat{x}$, $\hat{y}$, or $\hat{z}$ (red, green, and blue curves, respectively). Panel (a) is for vanishing SOC and panel (b) is for $\lambda_{SO}=0.1$. In the absence of SOC and magnetic field (black curve in Figure 5a), the Kondo peak suffers a splitting at ω=0 (see black curve in the inset for a detailed view). This resembles the physical behavior discussed in Ref. [38], where the resonant level was provided by the nearby non-interacting quantum dot coupled to a metallic lead. Here, it is provided by the propagating edge state of the ribbon, as observed in the black line of Figure 2b. A similar effect has been predicted for graphene zigzag nanoribbons [31]. In addition, at the vanishing SOC, ρ(ω) is the same for all three magnetic field orientations. In the main panel, it is possible to see that the $B_{z}$ result (blue curve) slightly diverges from the $B_{x}$ and $B_{y}$ results (red and green curves). This originates from the complexity of the Wilson chain discretization for the 2 × 2 matrix hybridization function that presents highly localized peaks, which somehow induce a small numerical error for a specific range of ω. For $\lambda_{SO}=0.1$, the energy bands around ω=0 become dispersive (see Figure 2a). This strongly suppresses the very narrow peak in the DOS observed at the vanishing SOC, making it much broader, resulting in an almost flat metallic DOS at the Fermi energy. As a consequence, the Kondo peak splitting, observed for $\lambda_{SO}=0$ in Figure 5a, caused by the very sharp peak in the DOS at the Fermi energy, is suppressed, as can be observed in more detail for the black curve (for *B* = 0) in the inset to Figure 5b. Obviously, this peak is split and suppressed once the magnetic field is switched on. In addition, for B≠0, because of SOC, ρ(ω) for $B_{x}$ and $B_{y}$ are identical (red and green curves), while it differs when the field is along the *z*-axis. This could be already inferred from the results shown in Figure 3b–d. This reflects the fact that the SOC effective magnetic field is along the *z*-axis (see Appendix B). It is also interesting to notice that the splitting and suppression of the Kondo peak, for finite SOC, is stronger when the field is applied in the xy-plane than when applied along the *z*-direction. As shown below, this seems to occur because the band is considerably more polarized when the field is applied in the xy-plane than when it is applied along the *z*-direction.

To investigate how the band polarization affects the Kondo state, an analysis of the impurity polarization $\langle S_{\alpha =x,y,z}^{\mathrm{imp}}\rangle$ dependence on the temperature and magnetic field is shown in Figure 6. We should remark that the impurity is polarized by the band; thus, at low temperatures, once the band polarization has completely destroyed the Kondo state, we should have a fully polarized impurity ($\langle S_{\alpha =x,y,z}^{\mathrm{imp}}\rangle ∼0.5$, at sufficiently high fields). Panel (a) shows (identical) results for $\langle S_{x}^{\mathrm{imp}}\rangle$ and $\langle S_{y}^{\mathrm{imp}}\rangle$, when the field is applied along the *x*- and *y*-direction, respectively, while panel (b) shows results for $\langle S_{z}^{\mathrm{imp}}\rangle$ when the field is applied along the *z*-direction. Obviously, the other $\langle S_{\alpha }^{\mathrm{imp}}\rangle$ components vanish. At high temperatures, as expected, all impurity magnetizations vanish for all values of field due to charge fluctuations, while, in panel (a), at low temperatures, the impurity polarization reaches a plateau, right below $\langle S_{x/y}^{\mathrm{imp}}\rangle ≲0.45$ for the highest field value. This value has increased monotonically with magnetic field from ∼0.4 (obtained for $\left|B\right|=10^{-4}$). A qualitatively similar picture is obtained for $\langle S_{z}^{\mathrm{imp}}\rangle$ [panel (b)], with the difference that the plateaus are lower and they are formed at somewhat lower temperatures, again indicating the fact that the band polarizes more strongly for magnetic fields in the xy-plane. Thus, as observed above for the different properties, the SOC-generated anisotropy is clearly observed.

## 3. Summary and Conclusions

In this work, we have investigated the Kondo effect in a system composed of a quantum magnetic impurity coupled to the edge of a ZSNR (in a hollow-site configuration). The ZSNR, a finite topological insulator, thus having a metallic topological edge state, is simulated by the Kane–Mele model, while the whole system is simulated by the SIAM. Our main interest is to simulate what an STM probe, analyzing the impurity, would see when the band is polarized by an external magnetic field applied along one of the coordinate axes. To emphasize the band polarization effect, we consider $g_{\mathrm{imp}}=0$, qualitatively similar to an experimental situation where the impurity gyromagnetic-factor is much smaller than the band gyromagnetic-factor, i.e., $g_{\mathrm{imp}}\ll g_{b}$, something that may be experimentally achieved by a judicious choice of the impurity. It turns out that the presence of the ZSNR intrinsic SOC, responsible for an effective magnetic field applied perpendicular to the ZSNR (*z*-direction), results, at small magnetic fields, in a considerably smaller band polarization than when SOC vanishes. In addition, the SOC-broken SU(2) symmetry results in band polarizations that, for $|B_{\alpha }=0.001|$, are related by $\langle S_{z}^{b}\rangle \;/\;\langle S_{x/y}^{b}\rangle ∼0.2$. This considerable difference greatly magnifies the difference in what an STM sees when analyzing the impurity Kondo peak for fields along different coordinate axes. Indeed, as expected, the Kondo peak is much more strongly suppressed for a magnetic field along the *x*- or *y*-axis than along the *z*-axis (see inset to Figure 6b). The same anisotropy and quantitative difference are observed in the impurity polarization $\langle S_{x/y}^{\mathrm{imp}}\rangle$ caused by the band polarization (compare panels (a) and (b) in Figure 4).

The importance of our results rests in the fact that recent advances in nanoribbon synthesis, STM sensitivity, and the STM ability to precisely place adsorbed magnetic impurities in surfaces and monolayers, allow us to propose the use of the Kondo effect, when its Kondo peak is STM-observed, to test the ability of the Kane–Mele model to describe the basic properties of ZSNRs. We hope that our results will stimulate experimental and theoretical groups alike to test our predictions.

## Figures and Tables

**Figure 1 nanomaterials-12-01480-f001:**
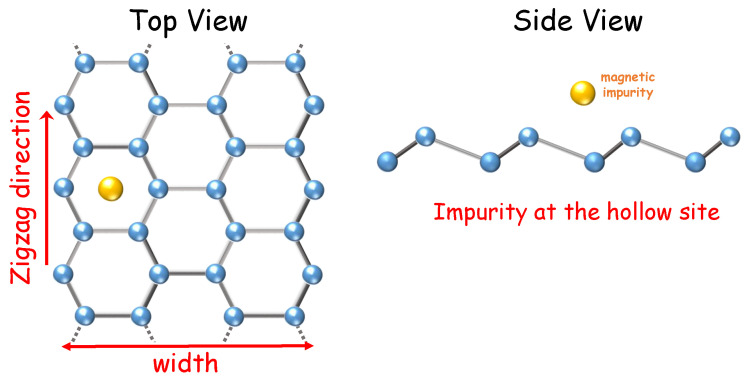
Schematic representation of a ZSNR (top view and side view) with a magnetic impurity located at the so-called hollow site.

**Figure 2 nanomaterials-12-01480-f002:**
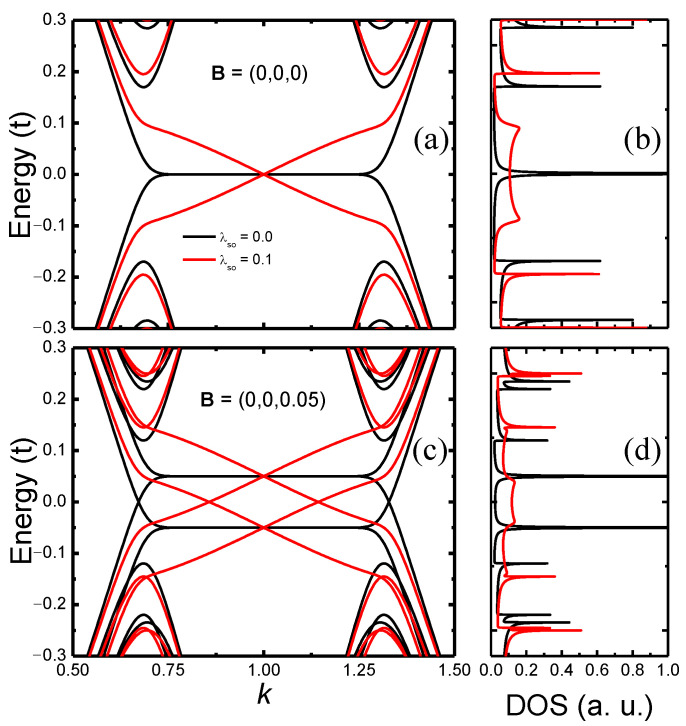
(**a**,**c**) N=26 ZSNR energy bands, as a function of momentum, and the respective total DOS, panels (**b**,**d**), for energies close to the Fermi level. Panel (**a**) presents results in the absence of magnetic field for two different sets of parameters [see legend in panel (**a**)], while panel (**c**) presents results for the same parameters as in panel (**a**), but with a magnetic field along the *z*-direction. Labels for each curve in all panels are displayed in (**a**).

**Figure 3 nanomaterials-12-01480-f003:**
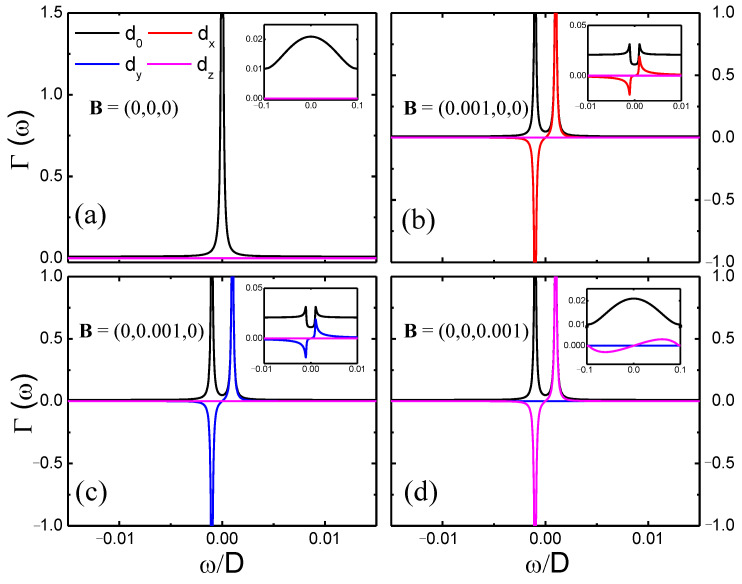
(**a**–**d**) Hybridization function coefficients $d_{\alpha =0,x,y,z}\left(\omega \right)$ for applied magnetic field along different directions, for an impurity placed at the hollow-site configuration (at the edge). The insets show the coefficients for $\lambda_{SO}=0.1$. For all panels N=26 and $V_{0}=0.05$.

**Figure 4 nanomaterials-12-01480-f004:**
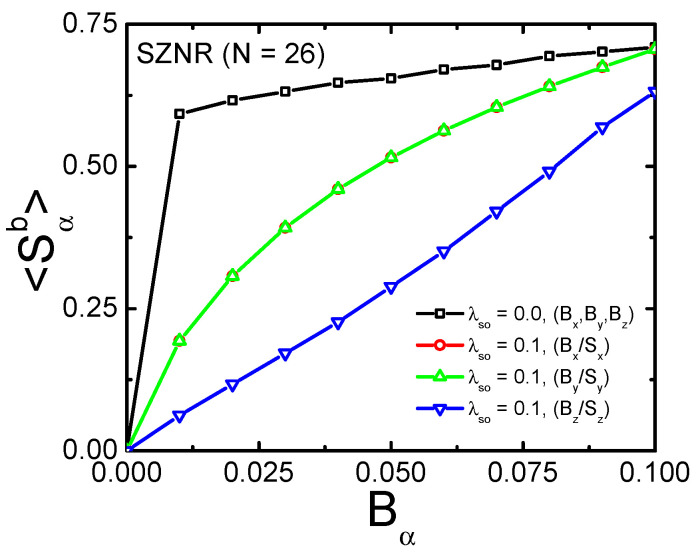
Band polarization $\langle S_{\alpha }^{b}\rangle$ for α=x,y,z, as a function of magnetic field applied along the coordinate axes. The black curve is for vanishing SOC (thus the results are independent of magnetic field orientation), while the red, green, and blue curves are for magnetic field applied along the *x*-, *y*-, and *z*-axis, respectively, for $\lambda_{SO}=0.1$.

**Figure 5 nanomaterials-12-01480-f005:**
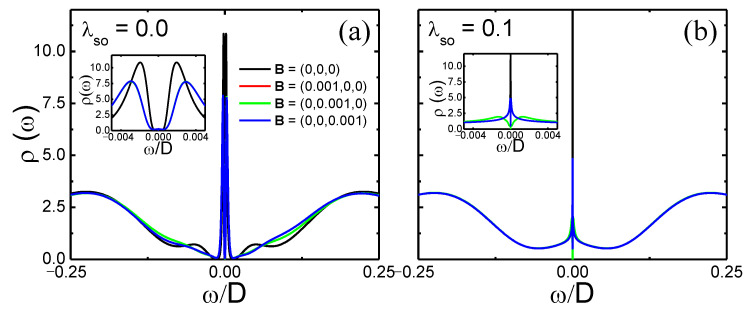
Impurity density of states, $\rho \left(\omega \right)=\rho_{↑}\left(\omega \right)+\rho_{↓}\left(\omega \right)$, for energies around the Fermi energy (ω=0). (**a**) Results for vanishing magnetic field (black curve) and for |B|=0.001 along the three coordinate axes (see legend), with $\lambda_{SO}=0.0$. (**b**) Same magnetic field values as in panel (**a**), but for $\lambda_{SO}=0.1$. The insets show zooms of the Kondo peak.

**Figure 6 nanomaterials-12-01480-f006:**
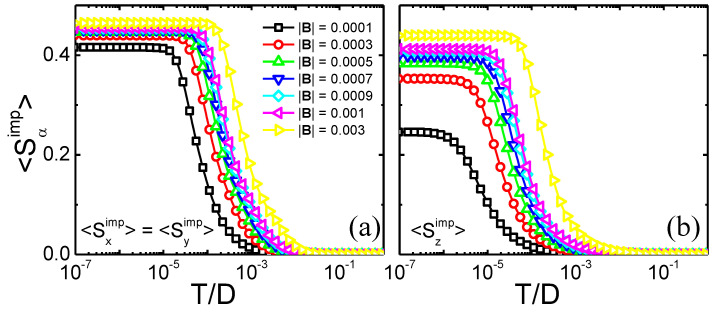
Impurity polarization $\langle S_{\alpha =x,y,z}^{\mathrm{imp}}\rangle$ as a function of temperature for different applied magnetic fields. (**a**) Magnetic field along *x*- and *y*-direction. (**b**) Same curves as in panel (**a**), but for magnetic field along the *z*-direction, with $\lambda_{SO}=0.1$ for both panels.

## Data Availability

Not applicable.

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
