# Peer review of "Band Polarization Effect on the Kondo State in a Zigzag Silicene Nanoribbon"

_nanomaterials, 2022, doi:10.3390/nano12091480_

Round 1

Reviewer 1 Report

Referee Report

Journal:  Nanomaterials

Manuscript ID:  Nanomaterials-1635004

Title:  Band polarization effect over the Kondo state in a zigzag silicene nanoribbon

Authors: G. S. Dinz, E. Vernek, and G. B. Martins

The authors discuss  impact of magnetic field induced  polarization of edges of zigzag silicene nanoribbon  on Kondo effect of magnetic impurity placed in a hollow-site configuration and  coupled to the edge. The tight binding model of a ribbon  with a single nearest neighbor hopping parameter is used and edge polarizations, as well as  hybridization functions dependent on the value and orientation of magnetic field are found. The numerical data indicate the spin-orbit introduced anisotropy.   The  influence of magnetic field  and spin-orbit coupling on the shape of many-body resonance and impurity polarization  is illustrated by the calculations performed by  Numerical Renormalization Group method.    

    In my opinion the  study adds new interesting  knowledge to the field and can serve as a hint in searching new potential application of graphene–like systems. The paper deserves for publication,  however, I have a few  remarks and questions that should be adequately addressed before publication.

1) Kondo physics strongly depends on the details of hybridization function and thus on the band structure of the nanoribbon. The authors use   a  crude tight binding approximation with only one hopping integral.  For purely model considerations it is sufficient.  However, the article is addressed  to a specific physical system – zigzag  silicene nanoribbon.  There are in literature  band structure calculations on  silicene ribbons  based on first principles methods (e.g.  Trivedi et al. , J. Comp. and Theor. Nanoscience 11, 781).  To make the results useful for experimentalists it would be extremely important to  check to what extent the simplified tight binding model agrees with ab initio calculations. The authors  should comment on whether the restriction to the n.n. hopping  is sufficient.

        2) Authors claim that assumption, that the impurity gyromagnetic factor is much smaller than the band gyromagnetic factor is experimentally achieved by a judicious choice of impurity. What  magnetic impurity have the authors in mind?

3) Magnetic field induces band polarization of edges. The authors include the impact of polarization by the field dependence of hybridization function (both on the value of the  field  and on its orientation).  It is well known, however, that cotunneling processes  involving polarized electrodes introduce also effective exchange splitting of Kondo resonance ( see e.g.  J. Martinek et al. Phys. Rev. B 72, 121302(R)).  I may be wrong, but if not, at least a comment regarding the neglect of this effect would be necessary

Author Response

Please, see reply on attached file.

Reviewer 2 Report

Dear Editor,

            The authors are investigated the properties of quantum impurity coupled to zigzag silicene nanoribbon that is subjected to the action of a magnetic field applied in a generic direction, using the numerical renormalization group method. In addition, a simulation of what a Scanning Tunneling Microscope will see when investigating the Kondo peak of a magnetic impurity coupled to the metallic edge of topologically non-trivial nanoribbon have been proposed.

The manuscript is well written, the results are solid and interesting for the specialists of the research field. I would like to recommend this manuscript for publication in the current form.

Author Response

Please, see reply on attached file. 

Reviewer 3 Report

Referee report on the manuscript by G.S.Diniz et al "Band Polarization Effect..."

The manuscript reports a theoretical study of silicene zigzag-like structure with magnetic impurity located near its edge. In fact, this is just one more paper in addition to a huge number of similar ones already published while there is still no experimental realization of silicene whose transport properties would be most interesting and important from the point of view of a possible practical application in silicon technology. In other words, the manuscript contains one more numerical study without any new deep and interesting physics. Nevertheless, it can still be published in Nanomaterials because the authors have shown sufficient competence in solving numerically Anderson-like Hamiltonian of the system under consideration.

Concrete remarks

  1. In the introduction the authors note: "For example, the fact that the conductance of the helical state in CdTe/HgTe was measured to be G < G0 = 2e2/h, and decreasing with lowering temperature, was attributed, among other factors, to the presence of either magnetic impurities coupled to the edges or to the presence of trapped electrons that interact with the helical state". This is wrong. Sometimes in 2DTI on the basis CdTe/HgTe structure G < G0 is due to a trivial reason - bulk leakage (see JETP Lett., v. 111, p. 121 (2020)), and no signature of magnetic impurities was found in HgTe based quantum wells (see Physics-Uspekhi v. 63, p. 629 (2020). So I advice to remove this statement from the paper.

  1. The authors declare that in silicene spin-orbit interaction (SOI) is much larger than in graphene. This statement is rather empty, because SOI of practically any strength above zero would be larger than that in graphene where it is very low. As a matter of fact, the value of SOI in silicene is unknown. And in the manuscript there is no data concerning it. But it is well-known that in 2D electron systems in silicon MOSFETs the effects caused by spin-orbit interaction are not observed (see Ando, Fowler and Stern Electron properties of 2D electron systems, Rev.Mod.Phys, 54, 437 (1982). The authors should specify what magnitude of SOI they expect in silicene.

Author Response

Please, see reply on attached file. 

Reviewer 4 Report

Diniz, Vernek, Martins:

Band Polarization Effect Over the Kondo State in a Zigzag Silicene Nanoribbon

Diniz et al. report on a detailed theoretical study of a magnetic quantum probe, i.e. a single magnetic atom embedded in a silicone nanoribbon. The quantum state of the magnetic atom in zero and finite external magnetic field is investigated by scanning tunneling microscopy. The magnetic atom is treated in terms of the single impurity Anderson model.

The theoretical part appears to be carefully performed and focusses on a problem current interest. However, in my feeling the critical model assumption, namely g_imp=0 or at least vanishing small, is largely unrealistic. Throughout the manuscript the authors assume a ‘magnetic impurity’ which at the end cannot respond to an external magnetic field. Apparently, the authors appear themselves somewhat uncomfortable with this assumption since in the Results and Discussion chapter they try to moderate the consequences of such an assumption by saying that the magnetic impurity can be selected by a ‘judicious choice’ to come close to an experimental realization. What ‘magnetic impurity’ with g_imp=0 do the authors have in mind?

In summary, certainly a sound theoretical treatment but largely of pure academic interest and with no reference to a realistic experimental setting. In view of this major disadvantage I recommend not to accept the manuscript for publication.

Author Response

Please, see reply on attached file. 
